# ACCURATE TOKENIZATION OF 3D SMALL ORGANIC MOLECULES

**Filya Geikyan**[1,2]**, Hrant Khachatrian**[1,2]
[1]YerevaNN Research Lab, Yerevan, Armenia
[2]Yerevan State University, Yerevan, Armenia

## ABSTRACT

Atom-level generation of chemical structures, including drug-like molecules, is an increasingly active research direction. Due to the continuous nature of atomic coordinates, 3D structure generation has been mostly done with diffusion-style methods, with only a few attempts at leveraging autoregressive models. In this work, we develop CoordToken, a simple recipe to train tokenizers for 3D molecules. We train CoordToken on two datasets: (i) on $\nabla^2$DFT where we obtain a $0.048$Å reconstruction error, which is a $4.2\times$ reduction compared to a prior method, and (ii) on a large corpus of 196M molecules, where we obtain micro averaged RMSD of $0.070$Å across all test datasets, including $0.074$Å on $\nabla^2$DFT. The tokenizer maintains near-perfect physical plausibility with a 98% pass rate on the PoseBusters checklist. The tokenizer and the corresponding dataset are available at `https://github.com/YerevaNN/CoordToken`.

## 1 INTRODUCTION

Large-scale databases of 3D chemical structures enabled a wave of research on generative models for designing new molecules and materials. As the coordinates are numeric, most of the work has been concentrated on methods based on diffusion and flow matching, to directly output the locations of atoms in the space Hassan et al. (2024); Wang et al. (2023). BindGPT Zholus et al. (2025) is the first autoregressive model that learns to generate atom coordinates in regular text format without specific tokenizers. Another attempt is presented in Bedrosian & Khachatrian (2025). Quetzal, introduced in Cheng et al. (2025), integrates a small diffusion decoder from Li et al. (2024) on top of an autoregressive model to avoid having a tokenizer. Gao et al. (2024) built a tokenizer trained on PCQM4Mv2 for each individual atom, by converting atom coordinates to spherical coordinates.

We believe the experience from image tokenizers in Chameleon Team (2024) (and, possibly, GPT-4o) shows that autoregressive models can handle continuous outputs well enough with discrete autoencoders. The closest work to ours is Bio2Token Liu et al. (2025). They have developed tokenizers for small molecules (`mol2token`), proteins and RNAs, and one for all modalities at once that can cover up to 100,000 atoms.

We collect a large and diverse corpus of 3D small-molecule structures and train CoordToken in two settings. First, on $\nabla^2$DFT (the same dataset used for `mol2token` Liu et al. (2025)), CoordToken reduces reconstruction error by $4.2\times$ (from 0.2 to 0.048Å). Second, training on the full collected corpus yields a $2.7\times$ reduction in reconstruction error on $\nabla^2$DFT (from 0.2 to 0.074Å), while maintaining high physical validity of the reconstructed molecules under PoseBusters-style checks. The second model achieves relatively consistent reconstruction error across diverse datasets. Further analysis shows that the tokenizer's error grows on implausible (noisy) molecular structures.

## 2 RELATED WORK

**Molecular Tokenization.** Tokenization plays a central role in adapting language-modeling techniques to molecules. Early approaches relied on SMILES strings Weininger (1988). Recent works such as Mol2Vec Jaeger et al. (2018), Mol2Context Lv et al. (2021), and `mol2token` Liu et al. (2025) have explored learned tokenizers that incorporate chemical and structural information into the

representation. These methods demonstrate that richer molecular tokenization can improve downstream generalization, but most focus on 1D or 2D representations.

**Molecular Generation.** Generative modeling of molecules has been extensively studied, with methods ranging from SMILES-based autoregressive models Olivecrona et al. (2017) to graph-based approaches Jin et al. (2018). Large-scale pretraining with GPT-style architectures has recently shown promise in capturing chemical grammar and enabling conditional molecule design Wang et al. (2019); Chilingaryan et al. (2024). Nevertheless, these approaches typically neglect explicit 3D structure, which is crucial for downstream tasks such as docking and quantum property prediction; incorporating geometry-aware representations can improve the physical plausibility of generated molecules Bedrosian & Khachatrian (2025).

**3D Molecular Representation Learning.** Several models learn directly from 3D coordinates, either by encoding molecules as point clouds Sch"utt et al. (2017) or by constructing equivariant graph neural networks Satorras et al. (2021). While these methods excel in prediction tasks, they typically rely on continuous embeddings, which can be memory-intensive and less suited for discrete compression or tokenization. More recently, discrete representation learning methods such as VQ-VAE van den Oord et al. (2017) and FSQ Mentzer et al. (2024) have been applied to molecular domains, enabling efficient latent codes for both generation and retrieval. Our work builds on this line by introducing a 3D-aware tokenizer that compresses per-atom coordinates into structured discrete codes, bridging the gap between sequence-based molecular tokenization and geometry-aware learning.

## 3  MODEL DESIGN

Our model, named CoordToken, is a transformer-based autoencoder for chemical structures with a finite-scalar quantization (FSQ) Mentzer et al. (2024) bottleneck. Both its encoder and decoder take atom types as input.

**Representing Molecules** Each molecule is represented as a sequence of per-token features derived from its SMILES string together with associated 3D coordinates. The tokenizer produces one discrete FSQ code per input token position (including non-atomic SMILES characters), aligned with the SMILES tokenization.

In preprocessing, we remove explicit hydrogens, generate RDKit canonical isomeric SMILES, align coordinates to atom tokens, and center each molecule before tokenization. Full preprocessing and tokenization details are provided in Appendix Section A.1.

We use a vocabulary of $V$ distinct SMILES tokens for all experiments. Each token is encoded as a one-hot vector over this vocabulary and concatenated with three real-valued coordinates. For tokens that do not correspond to atoms, the coordinate values are set to zero; we retain these non-atomic tokens because they might provide important structural information about molecular topology.

Sequences are padded to a maximum length of $T$ tokens, yielding inputs of shape $[B, T, F]$, where $F = V + 3$ and $B$ is the batch size. Across all experiments, we use the full-data vocabulary size $V = 192$. In the full-data setting, we set $T = 218$, which matches the maximum tokenized sequence length in the training corpus and therefore avoids truncation during training. For $\nabla^2$DFT experiments, we use $T = 54$. At inference time, we process each molecule at its native sequence length (without padding), and no truncation is applied.

**Encoder** The encoder projects per-token features into a $d_{\text{model}}$ dimensional embedding space, to which positional encodings are added. We use $d_{\text{model}} \in \{512, 1024\}$. The embedded sequence is processed by a stack of $L$ Transformer encoder layers, where we set $L = d_{\text{model}}/128$. Each layer uses multi-head self-attention with $h$ heads, where $h = d_{\text{model}}/64$ (so the head dimension is $d_{\text{head}} = d_{\text{model}}/h = 64$), and a feedforward sublayer of dimension $4d_{\text{model}}$.

**Quantization Bottleneck** The encoder projects its outputs into a $D_z$-dimensional continuous representation *per token position*, which the FSQ discretizes. We adopt the iFSQ refinement Lin et al. (2026) on top of FSQ, replacing FSQ's fixed activation mapping with a rescaled nonlinear mapping

that better distributes latent activations across quantization levels, improving code utilization and reconstruction quality. This provides a small empirical improvement (see Section 6).

We use $D_z \in \{12, 14\}$. For each token, each latent dimension is quantized to one of two values, producing $C = 2^{D_z}$ distinct discrete codes (e.g., $C = 2^{12} = 4096$ or $C = 2^{14} = 16384$); equivalently, each token is assigned a $D_z$-bit code under a base-2 quantizer. Unlike vector quantization in VQ-VAE, FSQ does not require a learned codebook; instead, it provides a structured and memory-efficient discrete representation that is well suited for per-token molecular compression.

While the codes are produced for every token position, the coordinate channels are meaningful only at atom tokens (non-atomic tokens have zeroed coordinates by construction). The decoder (and any downstream autoregressive model) can condition on atom types separately, alongside these coordinate codes.

**Decoder** The decoder input is constructed by concatenating the atom-type one-hot vectors with their corresponding quantized FSQ codes. This design ensures that the decoder has access both to atom identity and to the compressed coordinate representation. The concatenated vectors are projected back into the $d_{\mathrm{model}}$ space and passed through a Transformer stack structurally identical to the encoder. A final linear layer predicts the 3D coordinates for each token.

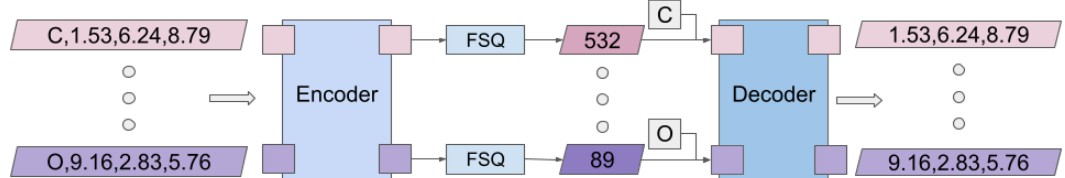

Figure 1: Overview of CoordToken architecture.

Figure 1 summarizes the overall training pipeline for CoordToken.

**Training Objective** The training loss is defined as the mean squared error (MSE) between predicted and true atomic coordinates. Because the input sequences contain both padding and non-atomic SMILES tokens, a mask is applied so that only valid atom positions contribute to the loss. Formally:

$$\mathcal{L} = \frac{1}{N_{\mathrm{atoms}}} \sum_{j=1}^{N_{\mathrm{mol}}} \sum_{i \in \mathcal{A}_j} \|\hat{\mathbf{x}}_i^{(j)} - \mathbf{x}_i^{(j)}\|_2^2, \quad \text{where} \quad N_{\mathrm{atoms}} = \sum_{j=1}^{N_{\mathrm{mol}}} |\mathcal{A}_j|. \tag{1}$$

where $N_{\mathrm{mol}}$ is the number of molecules in the batch and $\mathcal{A}_j$ is the set of valid atom positions for molecule $j$ (excluding both padding and non-atomic SMILES tokens). This corresponds to an atom-level MSE: molecules with more atoms contribute proportionally more to the loss.

**Reconstruction Metric for Evaluation** During evaluation, we compute the root-mean-square deviation (RMSD, in Å) between reconstructed and reference structures to measure 3D reconstruction accuracy. We report RMSD *without optimal rigid alignment/superposition* (i.e., without a Kabsch fit), because in many downstream applications, such as pocket-conditioned conformer generation, the molecule must be generated in a specific orientation. The tokenizer should therefore preserve orientation information rather than factor it out. For a molecule with $N$ atoms, we compute

$$\mathrm{RMSD} = \sqrt{\frac{1}{N} \sum_{i=1}^{N} \|\hat{\mathbf{x}}_i - \mathbf{x}_i\|_2^2}, \tag{2}$$

where $\hat{\mathbf{x}}_i \in \mathbb{R}^3$ and $\mathbf{x}_i \in \mathbb{R}^3$ denote the predicted and reference coordinates of atom $i$. We refer to this metric as RMSD throughout.

**Physical Validity Metric for Evaluation** In addition to RMSD, we report physical validity using PoseBusters-style checks Buttenschoen et al. (2024). We call a molecule valid if it passes all tests, and report accuracy as the fraction of molecules that are valid (i.e., # valid / # total).

**Optimization Details** We used the AdamW optimizer and trained using bf16 precision. For the small $d_{model} = 512$ ablation setting, the hyperparameter search over batch size, learning rate, and FSQ/iFSQ variants is reported in Appendix Table 4. We used the WSD Wen et al. (2024) learning-rate schedule. We also applied gradient clipping to stabilize training.

## 4  DATASETS

We built our training corpus by aggregating publicly available 3D small-molecule datasets. Specifically, we used the publicly released Flowr.root subset Cremer et al. (2025), ChEMBL3D Nikitin et al. (2025) (one lowest-energy conformer per molecule), GEOM Axelrod & Gómez-Bombarelli (2022) (up to 30 lowest-energy conformers per molecule, following common practice Frank et al. (2025)), and $\nabla^2$DFT Khrabrov et al. (2024). For these datasets, we parsed all structures available to us. For very large databases (ZINC and PubChem3D) Irwin et al. (2020); Bolton et al. (2011), we subsampled $10\%$ of the molecules we were able to parse. We then filtered out molecules containing isotopes, which were not covered across datasets. This isotope filter removed 342 molecules from KIBA-3D, 759 molecules from ChEMBL3D, and 244,216 molecules from PubChem3D.

Our split construction is designed to prevent canonical-SMILES overlap across train, validation and test splits, both within and across datasets, so that evaluation reflects true generalization. The exception is $\nabla^2$DFT, where overlap between the predefined splits is very high. The details of the split construction are provided in Appendix Section A.3. Appendix Table 3 summarizes the resulting split sizes and the original structure-generation protocol for each dataset.

We train CoordToken tokenizers in two settings: (i) on $\nabla^2$DFT and (ii) on the full aggregated corpus described above. For $\nabla^2$DFT, we evaluate final models on three held-out test splits: `test_conformations`, `test_scaffolds`, and `test_structures`, which assess generalization under different distribution shifts. When reporting a single $\nabla^2$DFT test RMSD, we use the average across these three splits, since their differences are negligible in our experiments.

## 5  MAIN RESULTS

### 5.1  RECONSTRUCTION

We evaluate CoordToken by reconstruction accuracy, measured using RMSD, as described in Section 3. Our main reported experiment uses iFSQ, a codebook of size $C = 2^{12}$, $d_{model} = 1024$, and batch size 1024 (128 per GPU on 8 GPUs), trained for 1 epoch with learning rate $10^{-4}$. This setting was chosen via a grid search run for 20% of a standard epoch, with per-GPU batch size in $\{64, 128, 256\}$, learning rate in $\{4 \cdot 10^{-5}, 10^{-4}, 4 \cdot 10^{-4}\}$, and two weight-decay modes: standard decay applied to all parameters, and selective decay applied only to weight matrices while excluding bias terms and normalization parameters. The selective setting performed best.

The micro-averaged and macro-averaged reconstruction errors across all test datasets are **0.070**Å and **0.091**Å respectively. We report the full results for each dataset in Appendix Table 3.

We evaluate physical validity using PoseBusters Buttenschoen et al. (2024): a molecule is considered valid if it passes all checks, and we report validity as the fraction of valid molecules. The molecules reconstructed by the tokenizer achieve **98%** validity, evaluated on 100 samples from each test set, closely matching the validity of the input molecules in these datasets.

We further evaluated on the out-of-distribution GEOM-XL dataset and obtained 0.326Å RMSD. As expected, RMSD on GEOM-XL generally increases with tokenized sequence length (Appendix Figure 4). Note that GEOM-XL contains more diverse and conformationally flexible molecules.

For $\nabla^2$DFT, we ran the grid search in the same way as for the full-data setting, but for one full epoch; in this case, standard (non-selective) weight decay was the winner. The ablation results below and Appendix Figure 2 both report validation RMSD. Separately, on the held-out $\nabla^2$DFT test splits, the $\nabla^2$DFT-only model performs better; nevertheless, we prioritize the full-corpus tokenizer for most use cases because it provides a more general representation across diverse molecular datasets.

## 5.2 PERFORMANCE ON PHYSICALLY IMPLAUSIBLE STRUCTURES

To stress-test physical plausibility, we take 100 molecules from each evaluation set in our full-data setting and corrupt only the 3D coordinates while keeping the atom/SMILES tokens fixed. Specifically, for every atom token position we first replace the $(x, y, z)$ coordinate channels with i.i.d. standard Gaussian samples ($\mathcal{N}(0, 1)$), yielding a fully random 3D geometry for the same underlying molecule. We then additionally consider a milder corruption where we add i.i.d. Gaussian noise with standard deviation $\sigma$ to the original coordinates (again, only at atom token positions). For each corrupted input, we run it through the tokenizer/decoder to obtain a reconstructed structure, and report RMSD between the reconstruction and the corrupted reference coordinates and physical validity. We report detailed results in Appendix Table 5 and Appendix Figure 3. Notably, PoseBusters accuracy can sometimes improve relative to the corrupted input after reconstruction.

## 5.3 ROBUSTNESS TO TOKEN-SEQUENCE PERTURBATIONS

We also probed how strongly the tokenizer depends on the input token sequence itself, using the same evaluation protocol as in the previous subsection, i.e., 100 molecules from each test dataset. For the canonical input sequence, the average reconstruction RMSD is 0.0784Å. We then considered three alternatives: using a randomized SMILES ordering for the same molecule, randomly shuffling the molecule's tokens, and replacing them with completely random tokens. Measuring the RMSD between each resulting reconstruction and the reconstruction obtained from the canonical input yields 0.0003Å, 0.1377Å, and 0.4704Å, respectively. The corresponding reconstruction RMSDs against the ground-truth structure are 0.0785Å, 0.1521Å, and 0.4718Å. Thus, changing to a different but chemically equivalent SMILES ordering has almost no effect, whereas disrupting or randomizing the token identities substantially degrades reconstruction. Taken together, these results show that CoordToken is not driven by coordinates alone: it uses molecular token identities to recover the underlying structure, while remaining nearly invariant to different valid SMILES orderings of the same molecule.

## 6 ABLATIONS ON $\nabla^2$DFT

Appendix Table 4 reports the small $\nabla^2$DFT hyperparameter search over batch size, learning rate, and FSQ/iFSQ variants ($d_{\text{model}} = 512$, 4 epochs), where iFSQ generally performs better than FSQ.

In the small-scale setting ($d_{\text{model}} = 512$, 4 epochs, batch size 128, learning rate $10^{-3}$), we assess the impact of atom-type conditioning and structural SMILES context: removing atom types from both encoder and decoder gives RMSD 0.149 (vs. 0.091 default), and removing them only from the decoder is even worse (RMSD 0.162). Restricting the tokenizer input to atom tokens with coordinates only, while dropping the other SMILES characters, further degrades reconstruction to RMSD 0.205. We therefore keep both atom-type conditioning and the full SMILES tokenization by default.

We next vary the tokenizer capacity by changing the number of FSQ codes ($C = 2^{D_z}$) and the autoencoder model dimension $d_{\text{model}}$. These runs are trained for 4 epochs. Appendix Table 2 reports RMSD on $\nabla^2$DFT. Increasing $d_{\text{model}}$ from 512 to 1024 substantially improves reconstruction. At fixed $d_{\text{model}}$, changing $C$ from $2^{12}$ to $2^{14}$ gives mixed gains across settings. We therefore keep $C = 2^{12}$ as the default for comparability across experiments.

## 7 CONCLUSION

We introduced CoordToken, a simple method for learning discrete tokenizers for 3D small molecules that compresses atomic coordinates into structured codes suitable for autoregressive modeling. Across both $\nabla^2$DFT and a large-scale pretraining corpus, CoordToken gives strong reconstruction accuracy while preserving near-perfect physical plausibility under PoseBusters-style checks. Generalization to larger molecules, as demonstrated by experiments on GEOM-XL is still challenging. The trained tokenizers, codebase and the datasets are publicly available to support future research in scalable, geometry-aware molecular generation.

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

## A APPENDIX

### A.1 PREPROCESSING AND TOKENIZATION DETAILS

In preprocessing, we remove explicit hydrogens and generate RDKit canonical isomeric SMILES. We then tokenize the canonical string with a rule-based SMILES tokenizer that treats atom symbols or bracket atoms as atom tokens, while representing bond, branch, ring-closure, dot, and stereochemical bond markers as separate structural tokens. Coordinates are aligned to atom tokens using

RDKit's _smilesAtomOutputOrder, so bracket atoms, aromatic atoms, and stereochemically annotated atoms each receive exactly one coordinate triplet, whereas the other tokens receive none. The final vocabulary is built from the resulting atom-with-coordinate tokens and structural SMILES tokens. For each molecule, we use a single RDKit canonical isomeric SMILES string (i.e., no random SMILES enumeration during training). Since different data sources use different coordinate ranges, we center each molecule by subtracting its center of mass, so that its coordinates are centered around the origin before tokenization. This removes global translation and places all molecules in a common centered frame, but it does not enforce rotational or reflection invariance. As a result, generated coordinates are produced in this centered frame (global placement can be applied afterward), and centering primarily improves cross-dataset generalization to translation/offset differences.

## A.2 Additional ablation tables

The main-text ablation references are reported here for completeness. The ablation tables below report validation RMSD on $\nabla^2$DFT.

Table 1: Effect of removing atom-type information or structural SMILES tokens on RMSD ($\nabla^2$DFT, 4 epochs).

| Description | RMSD |
|---|---|
| default | 0.091 |
| no atom types to decoder | 0.162 |
| no atom types to encoder and decoder | 0.149 |
| atom tokens with coordinates only | 0.205 |

Table 2: Tokenizer-capacity ablations on $\nabla^2$DFT.

| $C$ | $d_{\text{model}}$ | LR | RMSD $\downarrow$ |
|---|---|---|---|
| $2^{12}$ | 512 | $10^{-3}$ | 0.091 |
| $2^{14}$ | 512 | $10^{-3}$ | 0.094 |
| $2^{12}$ | 1024 | $10^{-4}$ | 0.069 |
| $2^{14}$ | 1024 | $10^{-4}$ | 0.061 |

## A.3 Dataset split construction

For clarity, we group datasets into three categories. **Group A** contains datasets with predefined splits: {BindingMOAD, BindingNet-High, BindingNet-Low, BindingNet-Mid, CrossDocked2020, DAVIS-3D, HiQBind, Kinodata-3D, OMol25-bio-mols, Plinder, SAIR, SPINDR, ChEMBL3D, GEOM}. **Group B** contains datasets without predefined splits: {KIBA-3D, OMol25-small-mols, ZINC, PubChem3D}. **Group C** contains {$\nabla^2$DFT}, which provides only train/test splits. We first processed Group A. We aggregated canonical SMILES from all available Group A test sets together with the GEOM-Revisited test split, then removed from all training and validation sets any samples whose SMILES overlapped with this aggregated test list. We included GEOM-Revisited at this stage because we may later evaluate downstream on that benchmark and wanted the evaluation to remain fair. Similarly, if a training split contained SMILES appearing in validation sets, those samples were removed from training.

Next, for Group B, we iteratively constructed new splits. For each dataset, we initially targeted test and validation sizes of roughly $0.5\%$ of the full dataset. Test candidates were selected as samples whose SMILES appeared in test sets of Group A and in test sets of previously processed Group B datasets; validation candidates were selected analogously. Training candidates were defined as samples whose canonical SMILES did not appear in test or validation sets of any other dataset. When a candidate split exceeded its target size, we randomly subsampled it. Otherwise, we filled the split from training candidates (assigning all samples with the same SMILES to the same split), while ensuring that added SMILES did not overlap with training sets of other datasets.

Finally, for Group C ($\nabla^2$DFT), we observed substantial train–test overlap in canonical SMILES: removing train/test overlaps would leave only 91,335 molecules in train out of 8,759,191. Therefore, for $\nabla^2$DFT we only enforced that its training split has no overlap with validation/test splits of Groups A+B. Because $\nabla^2$DFT has no predefined validation split, we constructed validation from its filtered training data to reach $0.5\%$ of the full $\nabla^2$DFT size.

Table 3: Details of the large corpus of molecules used in the training. Train/Test/Val columns show the number of conformers in the splits. Test RMSD is the performance of our best model trained on the full corpus.

| Dataset | Train | Val | Test | Test RMSD | Original Generation Method |
|---|---|---|---|---|---|
| BindingMOAD | 20,763 | 130 | 467 | 0.102 | Experimental structures (PDB crystallography) |
| BindingNet-High | 211,479 | 127 | 3,729 | 0.095 | Template-based comparative complex modeling |
| BindingNet-Low | 252,760 | 1,891 | 6,847 | 0.102 | Template-based comparative complex modeling |
| BindingNet-Mid | 144,831 | 327 | 3,312 | 0.098 | Template-based comparative complex modeling |
| CrossDocked2020 | 79,074 | 268 | 100 | 0.098 | Molecular docking (AutoDock/Vina-based) |
| DAVIS-3D | 129 | 7 | 225 | 0.115 | Experimental structures or docking |
| HiQBind | 24,863 | 48 | 279 | 0.107 | Manual curation + experimental structures |
| KIBA-3D | 263,514 | 1,539 | 1,468 | 0.089 | Experimental structures or docking |
| Kinodata-3D | 62,027 | 96 | 225 | 0.086 | Experimental structures + template-based docking |
| OMol25-bio-mols | 1,348,889 | 36 | 225 | 0.085 | DFT ($\omega$B97M-V/def2-TZVPD) + molecular simulation |
| OMol25-small-mols | 3,751,240 | 21,098 | 27,642 | 0.115 | DFT ($\omega$B97M-V/def2-TZVPD) + molecular simulation |
| Plinder | 123,402 | 367 | 949 | 0.099 | Curated experimental structures (curation + cleaning) |
| SAIR | 1,523,594 | 99 | 225 | 0.107 | Boltz-1x generated 3D complexes |
| SPINDR | 25,892 | 56 | 225 | 0.089 | Curated experimental complexes (filtered, standardized/refined) |
| ChEMBL3D | 1,442,219 | 182,421 | 37,065 | 0.083 | AIMNet2 neural network potentials |
| GEOM | 5,261,074 | 702,416 | 711,228 | 0.074 | GFN2-xTB; some refined with DFT ($\omega$B97X-D/def2-SVP) |
| $\nabla^2$DFT | 8,687,509 | 63,333 | 3,907,073 | 0.074 | DFT ($\omega$B97X-D/def2-SVP) |
| ZINC | 73,267,017 | 369,922 | 370,075 | 0.062 | Omega-based |
| PubChem3D | 93,723,972 | 473,209 | 473,316 | 0.056 | PubChem conformer pipeline |

## A.4 HYPERPARAMETER SEARCH GRID

Table 4: Grid search on $\nabla^2$DFT-only runs with $C = 2^{12}$, $d_{\text{model}} = 512$, and 4 epochs. Rows are batch size per GPU and columns are learning rates. We also report one average epoch time per batch size. Here, `big` denotes runs with RMSD $> 0.5$Å; these runs were also very unstable.

| | iFSQ | | | | | | FSQ | | | | |
|---|---|---|---|---|---|---|---|---|---|---|---|
| | $10^{-4}$ | $4 \cdot 10^{-4}$ | $10^{-3}$ | $2 \cdot 10^{-3}$ | $4 \cdot 10^{-3}$ | | $10^{-4}$ | $4 \cdot 10^{-4}$ | $10^{-3}$ | $2 \cdot 10^{-3}$ | $4 \cdot 10^{-3}$ |
| 64 | 0.098 | 0.090 | 0.120 | 0.109 | big | 64 | 0.099 | 0.103 | 0.101 | 0.140 | big |
| 128 | 0.132 | 0.093 | 0.091 | 0.110 | big | 128 | 0.140 | 0.101 | 0.098 | 0.111 | big |
| 256 | 0.176 | 0.107 | 0.116 | 0.163 | 0.183 | 256 | 0.202 | 0.110 | 0.152 | big | 0.186 |

| Batch size per GPU | 64 | 128 | 256 |
|---|---|---|---|
| Average epoch time | 11:22 | 05:51 | 03:01 |

Because epoch time differs substantially across batch sizes while RMSD differences are small, we chose batch size 128 for the subsequent small ablation runs and batch size 64 for the larger $\nabla^2$DFT-only runs.

## A.5 TRAINING-STEPS ANALYSIS ON $\nabla^2$DFT

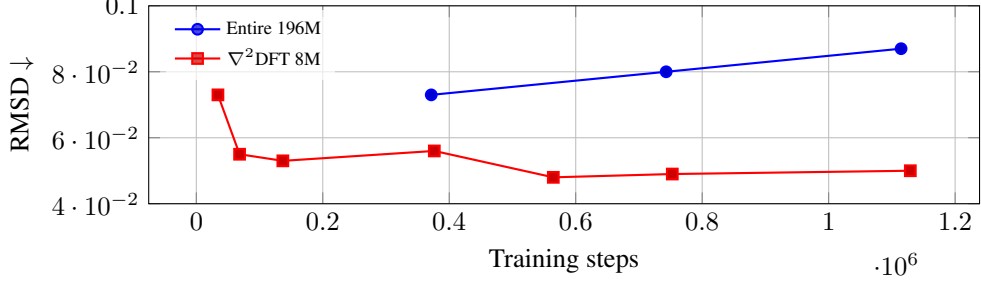

Figure 2: $\nabla^2$DFT validation RMSD versus training steps for models trained on $\nabla^2$DFT and on the full corpus; here, the full-corpus model degrades across epochs, while the $\nabla^2$DFT-only model improves overall.

## A.6 COORDINATE CORRUPTION VALIDITY

Table 5: RMSD and PoseBusters validity under coordinate corruption ("PB input": corrupted geometry; "PB output": reconstruction).

| Condition | RMSD (Å) ↓ | PB input acc ↑ | PB output acc ↑ |
|---|---|---|---|
| original | 0.078 | 0.982 | 0.981 |
| $\sigma = 0.01$ | 0.080 | 0.980 | 0.980 |
| $\sigma = 0.05$ | 0.106 | 0.980 | 0.981 |
| $\sigma = 0.1$ | 0.159 | 0.710 | 0.979 |
| $\sigma = 0.2$ | 0.284 | 0.041 | 0.979 |
| $\sigma = 0.4$ | 0.556 | 0.015 | 0.802 |
| random | 2.037 | 0.006 | 0.036 |

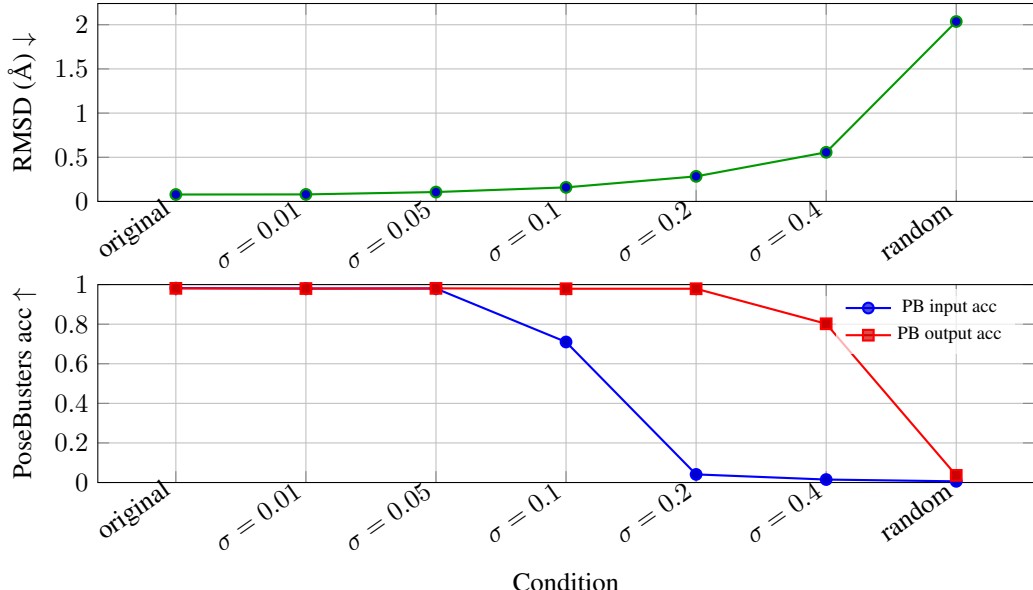

Figure 3: The performance of CoordToken on original and corrupted conformers. The conformer coordinates are corrupted by either adding noise with $\sigma$ standard deviation or by randomly sampling coordinates from $\mathcal{N}(0, 1)$. The bottom chart shows PoseBusters pass rates for each of the sets of conformers. The reconstructed conformers are at least as physically plausible as the corrupted ones.

## A.7 GEOM-XL SEQUENCE LENGTH ANALYSIS

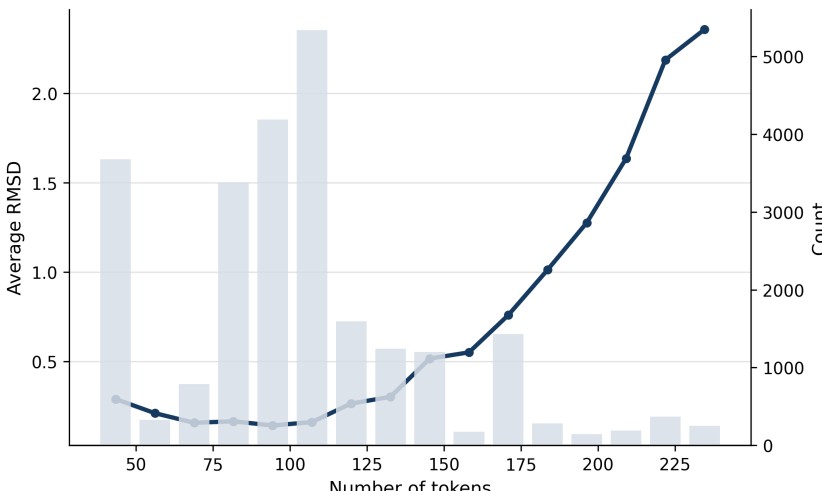

Figure 4: GEOM-XL reconstruction error versus tokenized sequence length. Longer tokenized sequences generally correspond to higher reconstruction error.

