# OpenReview forum: "Accurate Tokenization of 3D Small Organic Molecules"
_ICLR.cc/2026/Workshop/FM4Science — ICLR 2026 Workshop FM4Science Poster_

### Official Review · Reviewer_vCEK · 2026-02-20
**A nice work of using FSQ to solve the tokenization problem in 3D molecular representation and reconstruction**

**Rating:** 6
**Confidence:** 3

**Review:**

This paper introduces CoordToken, a Transformer-based autoencoder that utilizes Finite Scalar Quantization (FSQ) to compress continuous 3D atomic coordinates into discrete tokens. By training on publicly available 3D molecule datasets, the authors achieve a significant reduction in reconstruction error compared to previous SOTA methods while maintaining high physical plausibility. This work provides a critical bridge between geometry-aware molecular representations and the scalable power of autoregressive language models.
Strength:
1. Proposed a simple and efficient quantization framework for tokenization in 2D molecules
2. Experimental results verified that the model can achieve significant accuracy gains and a high validity rate

Questions:
1.  While the reconstruction results are impressive, how can we leverage this performance on other downstream tasks with generative goals?
2. Certain evaluation metrics may be too simple and only based on RMSD and physical validity; it is better to use diverse metrics for evaluation.
3. The training curve of Figure 2 only has 3 datapoints, which seems over-simplified.

---

### Official Review · Reviewer_No5U · 2026-02-22
**Strong Reconstruction Results, but Baseline Comparison Could Be Broadened**

**Rating:** 6
**Confidence:** 3

**Review:**

**Summary.** This paper introduces CoordToken, a transformer-based autoencoder that tokenizes 3D molecular coordinates using Finite Scalar Quantization (FSQ). The method is trained in two settings: on the $\nabla^2DFT$ benchmark and on a large corpus of 196M molecules. CoordToken achieves strong reconstruction accuracy while maintaining high physical validity under PoseBusters-style checks, and the authors commit to publicly releasing the trained tokenizers.

The contribution is concrete and clearly presented. Tokenizing 3D coordinates is a genuine bottleneck for autoregressive molecular generation, and the reported improvements over the prior method are substantial. The scope is appropriate for a workshop paper.

**Presentation issues.**

- The introduction contains a forward reference: "Section 4 summarizes the data sources used in this work." The paper would benefit from consistent cross-referencing in such cases.
- The iFSQ refinement (Lin et al., 2026) is used throughout but never explained beyond "provides a small empirical improvement." A one-sentence description of what it does would help readers unfamiliar with it.

**Questions for the authors.**

1. **Baseline comparison.** Several methods are discussed in the introduction and related work, including Gao et al. (2024) and Bedrosian & Khachatrian (2025), but the quantitative comparison is made exclusively against mol2token. Could the authors clarify why these other approaches were not included as baselines? It would help the paper to support the results with at least one more baseline.

2. **Out-of-distribution performance.** In Section 5.1, the GEOM-XL evaluation yields 0.298Å RMSD, roughly 6-7 times worse than in-distribution results. Could the authors discuss what drives this degradation and whether this is an expected result for more conformationally flexible molecules?

3. **Overfitting on $\nabla^2DFT$.** Figure 2 shows and the authors say that the $\nabla^2DFT$-only model worsens after 20 epochs. Could the authors comment on this overfitting and whether early stopping or regularization was considered?

4. **Physical validity phrasing.** Again, in Section 5.1, the paper states that reconstructed molecules achieve 93% validity, "matching the input molecules of the given datasets." Could the authors clarify whether this means the input molecules themselves have 93% validity under PoseBusters, and CoordToken preserves this rate?

**Strengths.** The method is simple, well-motivated, and clearly described. The ablations are thorough and directly address the key design choices. The questions above are relatively minor and the authors should be able to address them easily.

---

### Official Review · Reviewer_K6pM · 2026-02-22
**CoordToken reports strong reconstruction, but key evaluation choices and claims need tightening**

**Rating:** 4
**Confidence:** 4

**Review:**

This paper proposes CoordToken, an FSQ-based discrete tokenizer for 3D small-molecule coordinates aligned to SMILES tokens, trained at large scale (196M conformers) and evaluated via coordinate RMSD and PoseBusters validity. The reported reconstruction numbers are strong, but the evaluation setup (no rigid alignment, SMILES-aligned tokenisation, code allocation for non-atomic tokens) and baseline comparability need clearer justification, and the paper does not yet show impact on downstream autoregressive 3D generation.

CoordToken is a Transformer autoencoder with an FSQ/iFSQ bottleneck that produces a discrete code per SMILES token position (including non-atomic tokens whose coordinates are set to zero). The decoder conditions on atom types and predicts 3D coordinates. Training uses atom-masked MSE; evaluation reports RMSD without rigid alignment and PoseBusters-style physical validity.

Strengths?

Strong headline reconstruction: macro-average RMSD around 0.045 Å across many datasets; 0.044 Å on NablaDFT; OOD GEOM-XL is higher (0.298 Å) which is at least honestly reported.

Large-scale aggregation and reporting across diverse sources (experimental, docking, DFT, conformer pipelines) is useful, and the appendix table is informative.

Ablations are relevant (iFSQ, atom-type conditioning, code size, model size). The finding that atom types in the decoder matter a lot is important for downstream use.

The coordinate-corruption experiment is a reasonable stress test for “plausibility under noise” and shows the decoder can act as a denoiser/prior in some regimes.


Weaknesses and required clarifications?

RMSD without rigid alignment needs a stronger justification and makes comparisons risky. You explicitly report RMSD with no Kabsch fit. That can be valid for a tokenizer if coordinate frames are consistent, but then you must explain why frames are consistent across datasets and why this choice is comparable to prior baselines (many papers align). Right now the “3.5×” and “4.5×” improvement claims are hard to interpret without a strictly matched metric/protocol.

The tokenisation design is SMILES-aligned and assigns codes to non-atomic tokens, but the implications are not analysed. Non-atomic tokens have zero coordinates and do not contribute to the loss, yet you still generate codes for them. This can waste bitrate, create unconstrained codes, and complicate downstream autoregressive modeling. At minimum, quantify the overhead (bits per molecule) and show whether dropping non-atomic-token codes hurts.

Coordinate-frame and SMILES-canonicalisation assumptions are under-specified. You center by center of mass, but do not discuss rotation, reflection, multiple SMILES enumerations per molecule, or how you canonicalise/tokenise SMILES across datasets. For a discrete coordinate code that is meant to support generative modeling, these choices matter.

Physical validity reporting is not yet convincing as a main claim. You report ~93% validity, “matching the input molecules.” That reads as “no degradation,” which is fine, but it is not “near-perfect” unless inputs are near-perfect and you show per-dataset breakdowns where the tokenizer would realistically fail. PoseBusters is also not tailored to every small-molecule setting; justify its suitability here beyond “style checks.”

No demonstrated downstream value. The motivation is to enable autoregressive 3D generation, yet the paper only evaluates autoencoder reconstruction and validity checks. For acceptance, I would want at least a small downstream experiment: train a simple LM on CoordToken sequences and evaluate generated conformers with RMSD/validity, or show retrieval/compression tradeoffs that clearly outperform trivial quantisation baselines.

Presentation issues. The PDF header says “Under review at the GEM workshop, ICLR 2026” while the submission is to FM4Science. Fix template/venue text. Also make sure all symbols and units render cleanly (Å, dataset names).


What I would require for acceptance?

A clean, apples-to-apples baseline comparison where all methods use the same RMSD definition (with and without alignment), same preprocessing, and the same test splits.

A bitrate vs error analysis: report bits per atom and bits per molecule under your actual scheme (including non-atomic tokens), and compare to simple coordinate discretisation baselines.

A downstream demonstration (even small): an autoregressive model using CoordToken that produces 3D structures with reasonable validity and error, or a clear argument why reconstruction-only is sufficient for the workshop contribution.

Clarify canonical SMILES handling and coordinate-frame assumptions, and explain consequences for generative modeling and cross-dataset generalisation.


Questions for the authors?

Are RMSD numbers computed after any normalisation beyond COM centering (scaling, rotation canonicalisation)? If not, why is “no alignment” the right choice?

Are baseline numbers (Bio2Token / mol2token / others) computed with the exact same RMSD protocol? If not, please rerun baselines or restate the claim more cautiously.

Why emit codes for non-atomic SMILES tokens when their coordinates are zero and excluded from the loss? Did you test “atom-tokens only” coding?

What is the effective compression ratio (bytes per conformer) for typical molecules in your corpora?

Can you show at least one downstream AR generation result using CoordToken, even on a subset?

---

### Decision · Program_Chairs · 2026-03-03

Accept (Poster)